# Hepatitis E, Schistosomiasis and Echinococcosis–Prevalence in a Cohort of Pregnant Migrants in Germany and Their Influence on Fetal Growth Restriction

**DOI:** 10.3390/pathogens11010058

**Published:** 2022-01-03

**Authors:** Janine Zöllkau, Juliane Ankert, Mathias W. Pletz, Sasmita Mishra, Gregor Seliger, Silvia M. Lobmaier, Clarissa U. Prazeres Da Costa, Vera Seidel, Katharina von Weizsäcker, Alexandra Jablonka, Christian Dopfer, Michael Baier, Thomas Horvatits, Ingrid Reiter-Owona, Tanja Groten, Benjamin T. Schleenvoigt

**Affiliations:** 1Department of Obstetrics, Jena University Hospital, 07747 Jena, Germany; janine.zoellkau@med.uni-jena.de (J.Z.); tanja.groten@med.uni-jena.de (T.G.); 2Institute for Infectious Diseases and Infection Control, Jena University Hospital, 07747 Jena, Germany; juliane.ankert@med.uni-jena.de (J.A.); Mathias.pletz@med.uni-jena.de (M.W.P.); 3Department of Obstetrics and Gynaecology, Heidekreis Klinikum, 29664 Walsrode, Germany; Sasmita.Mishra@heidekreis-klinikum.de; 4Center for Reproductive Medicine & Andrology, Department of Obstetrics & Prenatal Medicine, University Hospital Halle (Saale), Martin Luther University Halle-Wittenberg, 06120 Halle, Germany; gregor.seliger@uk-halle.de; 5Department of Obstetrics and Gynecology, Klinikum Rechts der Isar, Technical University of Munich, 81675 Munich, Germany; silvia.lobmaier@tum.de; 6Center for Global Health, Institute for Medical Microbiology, Immunology and Hygiene, Technical University of Munich, 81675 Munich, Germany; clarissa.daCosta@tum.de; 7Clinic for Obstetrics, Charité, University Hospital, 13353 Berlin, Germany; vera.seidel@charite.de (V.S.); katharina.weizsaecker@charite.de (K.v.W.); 8Department of Rheumatology and Immunology, Hannover Medical School, 30625 Hannover, Germany; jablonka.alexandra@mh-hannover.de; 9German Center for Infection Research, Site Hannover-Brunswick, 38124 Hannover, Germany; 10Department of Pediatric Pneumology, Allergology and Neonatology, Hannover Medical School, 30625 Hannover, Germany; Dopfer.Christian@mh-hannover.de; 11Institute of Medical Microbiology, Jena University Hospital, 07747 Jena, Germany; michael.baier@med.uni-jena.de; 12Center for Internal Medicine, University Hospital Hamburg-Eppendorf, 20251 Hamburg, Germany; t.horvatits@uke.de; 13German Center for Infection Research (DZIF), Hamburg-Lübeck-Borstel and Heidelberg Partner Sites, 20359 Hamburg, Germany; 14Institue of Medical Microbiology, Immunology and Parasitology, University Hospital, 53127 Bonn, Germany; Ingrid.Reiter-Owona@ukb.uni-bonn.de

**Keywords:** hepatitis E, schistosomiasis, echinococcosis, pregnancy, migrants, fetal growth restriction

## Abstract

Background: Infections, as well as adverse birth outcomes, may be more frequent in migrant women. Schistosomiasis, echinococcosis, and hepatitis E virus (HEV) seropositivity are associated with the adverse pregnancy outcomes of fetal growth restriction and premature delivery. Methods: A cohort study of 82 pregnant women with a history of migration and corresponding delivery of newborns in Germany was conducted. Results: Overall, 9% of sera tested positive for anti-HEV IgG. None of the patients tested positive for anti-HEV IgM, schistosomiasis, or echinococcus serology. Birth weights were below the 10th percentile for gestational age in 8.5% of the neonates. No association between HEV serology and fetal growth restriction (FGR) frequency was found. Conclusions: In comparison to German baseline data, no increased risk for HEV exposure or serological signs of exposure against schistosomiasis or echinococcosis could be observed in pregnant migrants. An influence of the anti-HEV serology status on fetal growth restriction could not be found.

## 1. Introduction

The European Community (EU) is facing the greatest influx of refugees and migrants since the Second World War [1]. A total of 1.7 million people were estimated to be migrating to Europe in 2021, predominantly with African and Asian origin (25%, respectively) [2]. Migrants may more often have been exposed to helminthic or other infections and may be unaware of the potential impact these infections may have. In a migrant cohort in Germany, Dopfer et al. reported the proportion of women of childbearing age was 18% with a pregnancy frequency of 9.1 ± 0.8% [3]. Therefore, the question arises as to whether pregnant migrant women may be infected and if an observable influence on pregnancy can be detected.

Schistosomiasis is a helminthic infection, with about 249 million people each year requiring preventive antiparasitic therapy in 78 countries, and is associated with significant morbidity worldwide. Prolonged urogenital schistosomiasis is known as a risk factor for bladder cancer [4,5]. Schistosomiasis has an estimated prevalence of 10–20% in the Asian and African geographic regions of migrant origin [6,7,8]. In a Malawian cohort, reproductive tract affection by Schistosoma was described in about 60% of women with *S. haematobium* ova excretion in their urine [9]. Whilst schistosomiasis of the pregnant uterus is reported and placental schistosomiasis has been associated with stillbirth [10,11], no transplacental transmission has been observed. Occasional reports of placental schistosomiasis (i.e., detection of schistosomiasis eggs in placental tissue) exist, but have never been examined prospectively [11,12,13]. An association of schistosomiasis with premature delivery and low birth weight has been postulated [14,15,16,17], but further data are required. 

Echinococcosis is a neglected tropical disease. In the developed countries of the northern hemisphere, cystic echinococcosis is imported predominantly by migrants and has a very low incidence [18]. In Germany, 78 cases were reported in 2019 to the Robert Koch Institute. Countries with the highest frequency of origin among the affected migrants were Syria (11 cases), Bulgaria (9 cases), Turkey and Iraq with 7 cases, respectively. In 2019, only two cases were observed in the German population. Almost half of them (48.7%) were found in females [19]. The seroprevalence of echinococcosis in migrant populations, especially pregnant women and their effect on birth outcomes, has not yet been described in the literature. 

Hepatitis E is an infectious disease with worldwide occurrence, caused by the hepatitis E virus (HEV). Contaminated drinking water transmits HEV genotypes 1 and 2, and outbreaks are mostly associated with precarious hygienic conditions, while genotypes 3 and 4 are mainly found in industrialized countries and are spread zoonotically. The anti-HEV IgG seroprevalence is estimated to range between 2% and 38% in Middle Eastern and North African countries [20], whereas it averages around 17% in Germany [21]. A recent study from Germany with 604 refugees identified a similar seroprevalence of 20%. The average anti-HEV seropositivity was 38% in migrants from Africa [22]. A similar result was found in migrants from Ghana, with 34% anti-HEV IgG positivity in a Dutch study of 1198 participants with different ethnic backgrounds [23]. Recent data from Italy reported a seroprevalence of 53% among Africans in a very small study group of 81 migrants [24]. A review published in 2019 analyzed the HEV seropositivity prevalence among African pregnant women. Overall, the proportion was found to be 29%, with the highest rates in North Africa at 50% [25]. There is evidence in the literature that tropical hepatitis E (genotypes 1 and 2) in pregnancy is associated with adverse birth outcomes [26]. There are no publications studying the anti-HEV IgG and IgM seroprevalence rates in pregnant migrants. The impact of HEV seroprevalence in that population on birth outcomes, such as preterm birth, low birth weight, and fetal growth restrictions, has never been systematically investigated. 

The purpose of this research was the examination of the schistosomiasis, echinococcosis, and hepatitis E seropositivity prevalence and their association with the adverse pregnancy outcomes of premature delivery and fetal growth restriction (FGR). 

## 2. Results

### 2.1. Study Population

In total, 82 pregnant women with medical care in Germany and the corresponding newborns were included. Recruitment took place at five sites in Germany, including Jena (n = 35), Walsrode (n = 21), Halle (n = 13), Munich (n = 8), and Berlin (n = 5). Table 1 shows the demographic and migration characteristics of the cohort. 

### 2.2. Medical and Maternity History

Smoking was reported in 3.7% of the sample (additionally former smoking 1.2% and non-smoking 95.1%). No alcohol consumption was described in 95.2% of the sample, while monthly consumption or less was reported among 4.9%. Hypertension (1.2%), diabetes (3.7%), and previous anemia (8%; median hemoglobin, 6.98 mmol/L; IQR, 1.40) were evident medical conditions. A total of 75.6% of women reported one or more previous pregnancies (live births in 82%, miscarriages in 16%, and stillbirths in 2%).

### 2.3. Maternity Care Serology 

One woman reported being HIV positive. HIV testing was unknown in 53.7% and negative in 45.1% of the sample. No positive test result for hepatitis B or C was evident (hepatitis B negative result, 91.5% unknown/not analyzed, 8.5%; hepatitis C negative, 17.1% and unknown/not analyzed, 82.9%).

### 2.4. Pregnancy Conditions

A total of 8.5% of the women reported gestational diabetes. Pregnancy-induced hypertension was evident in 2.4% and pre-eclampsia in 1.2% of the patients. Medication during pregnancy consisted of magnesium (9.8%), methyldopa (4.9%), and others (26.8%). No use of acetylsalicylic acid or metoprolol was reported. 

### 2.5. Serological Results

Serology was performed for schistosomiasis in 82 samples and for Echinococcus and hepatitis E in 62 samples, respectively. All samples tested negative for schistosomiasis and Echinococcus. Six of the 62 (9.7%) sera tested positive for anti-HEV. According to ethnicity, 1/3 (33.3%) of African, 4/34 (11.8%) of Oriental Asian, 0/12 (0%) of Caucasian, and 1/13 (7.7%) of “other” ethnicities tested positive for anti-HEV IgG.

### 2.6. Perinatal/Neonatal Outcomes

The sex of the newborns was equally distributed (52% male, 48% female). Vaginal delivery occurred in 59.7% of the sample (spontaneously, 57.3%; assisted vaginal, 2.4%). C-section was necessary in 40.3% of cases (primary, 18.3; secondary, 22%). Newborns had a median gestational age at birth of 39.71 weeks (IQR 2.43), a median length of 51 cm (IQR 3.0), a median birth weight of 3318 g (IQR 623), and a head circumference of 35 cm (IQR 2.0). The median placental weight was 500 g (IQR 105.0). A total of 21% of newborns were admitted to the NICU. No neonatal death was reported. The median APGAR was 10 at 5 and 10 min. The arterial umbilical cord pH was reported as 7.29 (IQR 0.14). 

Preterm birth (<37 weeks) concerned five neonates (6.1%) with birth at a median gestational age of 32.3 weeks (IQR 6.5). Five neonates (6.1%) had low birth weight (<2500 g); four of them were simultaneously preterm born. Low birth weight infants had a median birth weight of 1700 g (IQR 1265) with a median percentile for birth weight of 35.0 (IQR 37.25), a birth height of 38.0 cm (IQR 40.00), and a head circumference of 42.5 cm (IQR 41.25). 

The 10th percentile of birth weight by gestational age was underrun by seven newborns (8.5%) without any of them having a preterm birth (<37 weeks). In birthweight below the 10^th^ percentile, a difference between the newborn´s gender of female (n = 2, 4.7%) or male (n = 5, 12.8%) was observed, but did not reach no significance (*p* = 0.25). 

### 2.7. Perinatal/Neonatal Outcomes in Dependency of Anti-HEV Serology

The examination of neither the birthweight nor the placenta/birthweight ratio for the anti-HEV IgG positive and negative subgroup revealed any significant differences, as displayed in Table 2.

HEV Serology in the five preterm birth cases was negative. Serology in the low birth weight outcome (<2500 g) was negative in 4/5 cases (in one case serology was not conducted). FGR, defined as birth weight below the percentile, was observed in 5.3% and 16.7% for the anti-HEV IgG seropositive and seronegative subgroups (*p* = 0.32), respectively. 

The examination of newborns’ length and head circumference for the anti-HEV IgG positive and negative subgroups revealed no differences. There were no differences between subgroups for APGAR values after 10 min. For APGAR values after 5 min, the median was 10 in the HEV negative group (range 7 to 10) and 9 in the anti-HEV positive group (range 5 to 10). The difference demonstrated statistical significance (*p* = 0.04). 

## 3. Discussion

A unique cohort of 82 pregnant migrant women seeking medical care in Germany—almost half from African (48.8%) and more than a quarter from Oriental Asian (28%) countries of origin, with different modes of transport—was serologically investigated. Although no positive serology for schistosomiasis or echinococcosis was found in the reported cohort of 82 pregnant migrants, anti-HEV IgG tested positive in 9.7% of cases. These data demonstrate that, in comparison to the German anti-HEV-IgG seroprevalence of 15–20%, the prevalence in the reported mothers was not increased [27]. These data demonstrate that there was no increased risk of exposure to HEV despite the escape to Germany with difficult hygienic conditions and despite the sometimes increased seroprevalence in the countries of origin of the pregnant refugees. 

According to ethnicity, 1/3 (33.3%) of African, 4/34 (11.8%) of Oriental Asian, 0/12 (0%) of Caucasian, and 1/13 (7.7%) of “other” ethnicities tested positive for anti-HEV IgG.

Both birth outcome and FGR did not differ significantly between previously HEV-exposed and non-exposed women. The FGR rate of 6.5% (4/62) is in line with the referenced birth collective of Jena University Hospital during the recruitment interval (March 2017 to September 2018) of 7.7%. The impression of 16.7% FGR in the anti-HEV positive group could not determine a statistical trend or significance due to the limited group size of only six hepatitis E positive women. The statistically significant difference found in the five-minute APGAR value is not of clinical relevance, while the median was 9 vs. 10 in anti-HEV positive or negative women, respectively. 

The 6 anti-HEV IgG positive women (ethnicities are as follows: 4, Oriental Asian; 1, African; 1, “other” ethnicity) showed no differences in origin, transportation method during migration, or other demographic or outcome parameters compared with the 56 seronegative women.

As shown in our previous work, the birth weight percentile outcome is influenced by height, the weight of the mother at delivery, migration transport modality, the count of previous pregnancies and previous births, pregnancy-induced and previous hypertension, preeclampsia, and previous diabetes [28]. There are no other publications addressing pregnancy and birth outcomes of migrant pregnant women in Germany. A systematic review including American and European (Sweden and Denmark) studies, published in 2017, investigated the newborn risk for adverse birth outcomes [29]. 

Birth outcome studies in migrant women are heterogeneous in results. A population-based Belgic study (>1.3 million births, 1998–2010) reported an increased rate of perinatal mortality but no LWB (<2500 g) in the migrant group. Sub-Saharan African mothers’ offspring showed an elevated risk for LBW in comparison to Belgian mothers’ offspring [30]. A systematic review from 2010 postulated a higher incidence of LBW in south-central Asians with migration history to the US and Europe. Sub-Saharan African, Latin American, and Caribbean women had a higher risk of delivering a LBW child in Europe [31]. 

Despite multicentric recruitment over 18 months, the sample size and, therefore, the generalizability of the result has to be considered as limited. Furthermore, it has to be mentioned that seroprevalence (IgG), and not present or recent infection (IgM), in association with adverse birth outcomes was addressed.

## 4. Materials and Methods

A prospectively cross-sectional cohort study was performed examining pregnant women with migration history and their corresponding newborns between March 2017 and September 2018. Country of origin was defined as endemic for schistosomiasis according to the World Health Organization (WHO) [32]. Inclusion criteria, therefore, were: (1) full-aged pregnant women, (2) written informed consent, and (3) origin from schistosomiasis-endemic countries. Exclusion criteria were: (1) structural fetal malformations, (2) chromosomal fetal aberrations, and (3) no singleton. Annotation: Any anamnestic placental pathology, as well as further medical conditions with a potential effect on fetal growth, lead to exclusion.

### 4.1. Study Assessments

Demographics, age, and medical history were collected using a patient questionnaire in participants’ origin language. A standardized case report form (CRF) was used for data collection, and pseudonymization took place at the source. Items were: maternal height and weight, the country of origin, ethnic origin, smoking and alcohol consumption, medical conditions or medication, fetal gender, and pregnancy history, including gravidity and parity status as well as gestational age (by ultrasound or last menstrual period). Laboratory values were noted, if available, from maternity records or the clinical routine for the following: hemoglobin, eosinophils, HIV, hepatitis B (Hbs-Ag) and C status (Anti-HCV). After birth, primary (birth weight percentiles) and secondary pregnancy outcomes (FGR, stillbirth, and premature delivery) were assessed. Calculation of birth percentiles for height, weight, and head circumference were performed according to Voigt et al. [33]. Blood was drawn for serologically testing. A follow-up of newborns or mothers was not performed.

### 4.2. Ethical Statement

The Ethical Committee of the University Hospital Jena, Germany reviewed and approved the study (approval # 4629-12/15). Study sites in Berlin, Munich, Halle, and Walsrode obtained concomitant votes. An Informed Consent Form in participants’ native language was signed by each woman, permitting data and serum sample use for scientific purposes. 

### 4.3. Statistics

The statistical analysis was performed using SPSS Statistics Version 25. First, normal distribution assessment was completed using a Kolmogorov–Smirnov test. For normally distributed metric data and independent variables, a *t*-test was performed. For non-normally distributed metric data, a Mann–Whitney U-test was used. Nominal or ordinal data was analyzed using a χ^2^ test. Fisher’s exact test was performed for contingency tables. Significance was considered at a *p*-value < 0.05 (*). 

### 4.4. Schistosomiasis Serology

Data collection originally aimed at the evaluation of association between schistosomiasis seropositivity and adverse pregnancy outcomes. The detection of specific antibodies in serum samples was performed using a commercial ELISA assay (SCIMEDX, Denville, NJ, USA). Tests were conducted according to the manufacturer’s instructions. The entire study cohort (n = 82) showed negative schistosomiasis serology.

### 4.5. Hepatitis E Serology

Serological testing for anti-HEV IgG and anti-HEV IgM antibodies was completed using the commercially available WANTAI enzyme-linked immunosorbent assay (ELISA; Wantai, Beijing, China). The assay was performed according to the manufacturer’s recommendations. Absorbance was measured using a Microplate Reader (Dynatech MRX Microplate Reader, Dynatech Laboratories, Texas City, TX, USA) at 450 nm and 450/630 nm. Three serum samples underwent relevant hemolysis, inhibiting the serological anti-HEV testing; thus, anti-HEV results are only available for 62 patients.

### 4.6. Echinococcus Serology

For the serological diagnosis of echinococcosis, an enzyme-linked immunosorbent assay (Euroimmun, Anti-*Echinococcus*-ELISA IgG, reactivity Ratio >1.0 positive, Lübeck, Germany) was used for screening, followed by a Western Blot (LDBIO Diagnostics, Echinococcus Westernblot IgG, Lübeck, Germany) for the confirmation of ELISA positive or borderline samples. A total of 62 samples were tested.

## 5. Conclusions

In comparison to German reference data, no elevated anti-HEV seroprevalence was observed in pregnant patients with a migration background, while testing for schistosomiasis and echinococcosis serology was completely negative. No influence of the serology status on FGR or birth outcome was found. Advanced serologic testing in relation to migration should be considered in prenatal care. The risk of migrants acquiring HEV infection, schistosomiasis, or echinococcosis during the escape to Germany should not be overestimated.

## Figures and Tables

**Table 1 pathogens-11-00058-t001:** Demographic and migration characteristics.

Number of cases	N = 82	
Age [years; median (IQR)]	27 (11.0)	
Weight [kg; median (IQR)]	67.5 (25.0)	
BMI [kg/m^2^; median (IQR)]	24.42 (7.89)	
Year of migration to Europe; median (IQR)	2015 (2.0)	
Ethnicity [n (%)]	Oriental Asian	40 (48.8)
	African	23 (28.0)
	Caucasian	14 (17.1)
	Other	5 (6.1)
Most frequent countries of origin [n (%)]	Syria	29 (35.4)
	Somalia	10 (12.2)
	Nigeria	9 (11.0)
Transportation during migration [n (%)] (multiple answers possible)	By car	16 (19.5)
	By foot	20 (24.4)
	By airplane	39 (47.6)
	By boat	22 (26.8)
	By train	22 (26.8)

**Table 2 pathogens-11-00058-t002:** Characteristics and perinatal outcomes in dependency of hepatitis E serology (referring to anti-HEV IgG positivity).

	Anti-HEV Negative 56/62 (90.3%)	Anti-HEV Positive 6/62 (9.7%)	*p*-Value
child birthweight in g median (range)	3285 (1075–4445)	3404 (3040–4450)	0.23
birth length in cm median (range)	51 (37–56)	52 (50–55)	0.13
head circumference in cm median (range)	35 (27.5–38)	34 (32–37)	0.69
FGR (<10th percentile)	3/56 (5.3%)	1/6 (16.7%)	0.32
birthweight placenta in g median (range)	480 (250–850)	482 (350–620)	0.99
Placenta/birthweight ratio median (range)	0.15 (0.09–0.30)	0.14 (0.09–0.16)	0.20
APGAR 5′ median (range)	10 (7–10)	9 (5–10)	0.04
APGAR 10′ median (range)	8 (8–10)	8 (8–10)	0.19

## Data Availability

The data presented in this study are available upon reasonable request from the corresponding author. The data are not publicly available for patient privacy reasons.

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
