# Peer review of "Hepatitis E, Schistosomiasis and Echinococcosis–Prevalence in a Cohort of Pregnant Migrants in Germany and Their Influence on Fetal Growth Restriction"

_pathogens, 2022, doi:10.3390/pathogens11010058_

Round 1

Reviewer 1 Report

This is a well-described study about the real-world prevalence of Schistosomiasis, This is a well-described study about the real-world prevalence of Schistosomiasis, Echinococcosis, and hepatitis E in German immigrants. However, there are also some issues that the authors need to address:

Introduction:

Line 90: IgG and IgM seroprevalence rates in in pregnant migrants.

  • IgG and IgM seroprevalence rates in pregnant migrants.

Materials and Methods:

Results

Table 1: by boot: Is 'by boat' misspelled as 'by boot'? Please clarify the meaning of this line.

Line 106: previous anemia (8 %, median hemoglobin 6.98 mmol/l, IQR 1,40)

  • Check that the number after the IQR is used correctly.

Line 110: One women was reported HIV positiv.

  • One woman was reported HIV positive.

Line 124: 1/3 (33,3%) african, 4/34 (11,8%) oriental asian, 0/12 (0%) caucasian

  • Use capitalization correctly. (African, Asian, Caucasian)

Table 2.

FGR  line: 3/56 (5,3 %)

  • Please check that the punctuation is correct.

Placenta birthwight ratio line: 0,15 (0.09-0.30)

  • Please check that the punctuation is correct.

Line 128: C-sectio was 128 necessary

  • C-section was 128 necessary

Discussion: Appropriate

Line 169-170 phrase seems to be unnecessary. It would be good to add a description of these numeric results to the result section.

Line 191: but no LWB (< 2,500g) in the migrant groups.

  • but no LBW (< 2,500g) in the migrant groups.

Line 200: Inclusion criteria thereforewere: 1) pregnant women >18

  • Inclusion criteria therefore were: 1) pregnant women >18

Author Response

This is a well-described study about the real-world prevalence of Schistosomiasis, This is a well-described study about the real-world prevalence of Schistosomiasis, Echinococcosis, and hepatitis E in German immigrants. However, there are also some issues that the authors need to address:

  • Thank you for the kindly valuation of the general aspects and key findings within our report. All detailed listed issues were addressed and changed in the revised manuscript as listed below.
  • English language and style are fine/minor spell check was performed.

Introduction:

Line 90: IgG and IgM seroprevalence rates in in pregnant migrants.

  • One “in” deleted.

Materials and Methods:

Results

Table 1: by boot: Is 'by boat' misspelled as 'by boot'? Please clarify the meaning of this line.

  • Spelling corrected (marked red).

Line 106: previous anemia (8 %, median hemoglobin 6.98 mmol/l, IQR 1,40)

  • Check that the number after the IQR is used correctly.
  • Spelling corrected (marked red).

Line 110: One women was reported HIV positiv.

  • One woman was reported HIV positive.
  • Spelling corrected (marked red).

Line 124: 1/3 (33,3%) african, 4/34 (11,8%) oriental asian, 0/12 (0%) caucasian

  • Use capitalization correctly. (African, Asian, Caucasian)
  • Spelling corrected (marked red).

Table 2.

FGR  line: 3/56 (5,3 %)

  • Please check that the punctuation is correct.
  • Spelling corrected (marked red).

Placenta birthwight ratio line: 0,15 (0.09-0.30)

  • Please check that the punctuation is correct.
  • Spelling corrected (marked red).

Line 128: C-sectio was 128 necessary

  • Spelling corrected (marked red).

Discussion: Appropriate

Line 169-170 phrase seems to be unnecessary. It would be good to add a description of these numeric results to the result section.

  • Numeric result were added to precise the message.

Line 191: but no LWB (< 2,500g) in the migrant groups.

  • but no LBW (< 2,500g) in the migrant groups.
  • Spelling corrected (marked red).

Line 200: Inclusion criteria thereforewere: 1) pregnant women >18

  • Inclusion criteria therefore were: 1) pregnant women >18
  • Spelling corrected (marked red).

Reviewer 2 Report

The authors studied if the fact to be a migrant woman may have an increased risk of adverse birth outcomes and infections. A cohort study of 82 pregnant migrants attending medical care in Germany and the corresponding newborns was considered. 9% of sera tested positive for anti-HEV IgG. None of the patients tested positive for anti-HEV IgM, schistosomiasis, or echinococcus serology. They concluded that, compared to German baseline data, no increased risk for HEV exposure or serological signs of exposure against schistosomiasis or echinococcosis could be observed in pregnant migrants.

As the authors said, birth weight percentile outcome is influenced by mother’s height, mother’s weight at delivery, transport modality to Europe, number of previous pregnancies and number of previous birth, pregnancy-induced and previous hypertension, preeclampsia, and previous diabetes.

The condition of FGR could have many reasons (maternal, fetal and placental) that may explain this pathological condition. 

The study is very well conducted, and quite well written, and finds a bibliographic precedent, but I think it was difficult to find a clear correlation between these two aspects, given the various confounding factors.
In any case, although not considering it innovative, I believe it can convey a useful message.

Author Response

  • Thank you for the kind valuation of the general aspects and key findings within our report. Indeed, low incidence of seropositivity and the magnitude of potential confounding factors are limitations, but also a reportable finding not to overestimate the issue of seldom infection in this population or their association to adverse pregnancy outcome.

  • English language and style are fine/minor spell check was performed.

Reviewer 3 Report

In the present manuscript, the authors have studied the seroprevalence of schistosomiasis, echinococcosis and hepatitis E virus in pregnant migrants in Germany and found no association with fetal growth restriction. However, seroprevalence (IgG) provides evidence of past infections. How can seropositivity for one pathogen be linked to differences in foetal growth? Wouldn’t it be more appropriate to determine the prevalence of the pathogen or IgM seroprevalence? Moreover, the sample size used in this study is very small and many other factors could have influenced foetal growth. I understand that it is difficult to study larger cohort but it is very difficult to find any association and then make any conclusion with such limited sample size.

Author Response

  • Thank you for the kind valuation of the general aspects and key findings within our report. Indeed, limitation of sample size in research including pregnant migrant women in combination with low incidence of seropositivity is not only a limitation, but also a reportable finding not to overestimate the issue of seldom infection in this population or their association to adverse pregnancy outcome.

  • English language and style are fine/minor spell check was performed.

Round 2

Reviewer 3 Report

Thanks to the authors for providing an answer to my previous comments. I understand that the sample size is limited in research including pregnant migrant women and I agree that the finding not to overestimate the issue of seldom infection in this population is interesting. However, I think the limitations of this study (due to small sample size) should be discussed in more details in the discussion. Moreover, you should discuss why you investigated seroprevalence (IgG) and not present or recent infection (IgM) to study an association with adverse birth outcomes.

Author Response

We´d like to thank the reviewer again for providing a valuable comment regarding the limitations and discussion. Both aspects were included within the discussion section and outlined as limitations.